# Toward Concurrent Identification of Human Activities with a Single Unifying Neural Network Classification: First Step

**DOI:** 10.3390/s24144542

**Published:** 2024-07-13

**Authors:** Andrew Smith, Musa Azeem, Chrisogonas O. Odhiambo, Pamela J. Wright, Hanim E. Diktas, Spencer Upton, Corby K. Martin, Brett Froeliger, Cynthia F. Corbett, Homayoun Valafar

**Affiliations:** 1Department of Computer Science and Engineering, University of South Carolina, Columbia, SC 29208, USA; andrewsmith@sc.edu (A.S.); mmazeem@email.sc.edu (M.A.); odhiambo@email.sc.edu (C.O.O.); 2Advancing Chronic Care Outcomes through Research and iNnovation (ACORN) Center, Department of Biobehavioral Health & Nursing Science, College of Nursing, University of South Carolina, Columbia, SC 29208, USA; wrightpamelaj@sc.edu (P.J.W.); corbett@sc.edu (C.F.C.); 3Pennington Biomedical Research Center, Louisiana State University System, Baton Rouge, LA 70808, USA; hanim.diktas@pbrc.edu (H.E.D.); corby.martin@pbrc.edu (C.K.M.); 4Department of Psychiatry, Psychological Sciences, and Cognitive Neuroscience Systems Core Facility, University of Missouri, Columbia, MO 65211, USA; uptons@missouri.edu (S.U.); froeligerb@health.missouri.edu (B.F.)

**Keywords:** machine learning, neural networks, human activity recognition, smart healthcare, ecological momentary assessment, context-aware environments, bite detection, wearable sensors

## Abstract

The characterization of human behavior in real-world contexts is critical for developing a comprehensive model of human health. Recent technological advancements have enabled wearables and sensors to passively and unobtrusively record and presumably quantify human behavior. Better understanding human activities in unobtrusive and passive ways is an indispensable tool in understanding the relationship between behavioral determinants of health and diseases. Adult individuals (N = 60) emulated the behaviors of smoking, exercising, eating, and medication (pill) taking in a laboratory setting while equipped with smartwatches that captured accelerometer data. The collected data underwent expert annotation and was used to train a deep neural network integrating convolutional and long short-term memory architectures to effectively segment time series into discrete activities. An average macro-*F*_1_ score of at least 85.1 resulted from a rigorous leave-one-subject-out cross-validation procedure conducted across participants. The score indicates the method’s high performance and potential for real-world applications, such as identifying health behaviors and informing strategies to influence health. Collectively, we demonstrated the potential of AI and its contributing role to healthcare during the early phases of diagnosis, prognosis, and/or intervention. From predictive analytics to personalized treatment plans, AI has the potential to assist healthcare professionals in making informed decisions, leading to more efficient and tailored patient care.

## 1. Introduction

Over the past 30 years, research has shown that health is influenced by a complex interplay of biological, behavioral, and environmental factors. Despite widespread technological advancements, healthcare has been slow to adopt these innovations. This study explores the potential of using existing, commercially available wearable sensors (like smartwatches) to promote healthy behaviors such as weight management, exercise, and chronic disease self-management. There is a huge potential in adapting smart technologies, such as phones and watches, to develop more effective health-promoting interventions for behaviors such as weight loss, physical activity, and chronic condition self-management. The successful adoption of these devices to promote healthier behavior requires solving the crucial problem of characterizing and monitoring human behavior in a way that will be the most useful, unobtrusive, and personally relevant. When this is sufficiently realized, the subsequent step of developing the optimal intervention mechanisms and personalized interventions can be explored.

Despite changes in the regulatory environment around nicotine use, and the rapid uptake in electronic nicotine delivery systems, cigarette smoking remains the leading preventable cause of death and disability in the US and costs nearly USD 600 billion (about USD 1860 per person in the US) each year [1]. Although 12% of adults in the USA currently smoke [1], the majority want to quit, and among those that make a quit attempt, the majority relapse. Both internal (e.g., stress and craving) and external environmental contextual factors (e.g., drug-related cues) directly reinforce the ongoing maintenance of nicotine use. Using well-controlled laboratory practices to characterize smoking biobehavioral pathophysiology does not lend itself to collecting and measuring ecologically valid smoking behavior in the natural environment, thus resulting in the reliance on remote self-report assessments to characterize behavior outside of the laboratory. Therefore, leveraging technology to accurately and passively detect smoking behavior in real-time would provide a significant improvement in measurement fidelity, help to characterize the spatial and temporal contexts surrounding smoking behaviors for an individual, and thus may inform personalized interventions for smoking cessation.

In addition to smoking, increasing obesity rates and associated chronic diseases are largely attributed to eating behaviors and contextual factors [2,3]. Given that most dietary assessment tools rely extensively on self-reports and have limitations, including a high user and experimenter burden and underreporting [4,5], it is challenging to measure eating behaviors and concurrent predictors accurately and regularly (e.g., daily). With the advancement of technology, it is now possible to remotely capture and even intervene in dietary behaviors when they occur. The passive detection of eating and drinking events allows researchers to automatically prompt participants to capture details about their food intake and other contextual factors and objectively quantify adherence to dietary interventions [6,7], particularly those that limit when food and beverages can be consumed [8]. Thus, this ability has numerous applications, including increasing the completeness and accuracy of self-reported food intake data while minimizing participant and study staff burden and quantifying adherence to dietary interventions. Previous validation efforts focused on detecting specific behaviors (e.g., smoking or eating and drinking) in settings where similar but competing behaviors were minimal [9,10,11]. The current study, however, tested the validity of a gesture detection model more rigorously since similar competing behaviors were present.

In 2018, 51.8% of US adults were diagnosed with one chronic health condition, and 27.2% had two or more chronic health conditions [12]. Adherence to prescribed medications is associated with improved clinical outcomes for chronic disease management and reduced mortality from chronic conditions [13]. Medication adherence, the degree to which a patient’s medication taking behavior corresponds with the agreed upon recommendations from a healthcare provider [14], is a global public health challenge, as only about 45% of patients take their medication as prescribed [15]. As such, the estimated costs of morbidity and mortality from medication nonadherence is more than USD 528 billion annually [16]. Whereas strategies exist that remind patients to take their medication [17,18], practical mechanisms to determine if the medication was taken after the reminder remain elusive. To date, methods to measure medication adherence are either expensive and impractical, such as metabolite testing [19], are merely approximate due to proxy data such as pill counts or self-report [20].

Human activity recognition (HAR) has been treated as a crucial application area for wearable sensor technology, enabling detailed monitoring and analysis of various physical activities [21]. To address the limitations of multi-sensor dependence and achieve superior performance metrics, we focus exclusively on accelerometer data, thus contributing to more efficient and cost-effective HAR systems. Several works have presented methods for human activity recognition using only accelerometer data; however, the majority either apply their method only to simple locomotion activities, collect their data with something other than a wrist-worn wearable, or use hand-crafted features instead of end-to-end deep learning [22,23,24,25]. Other research efforts have leveraged sensor data from smartwatches for HAR; however, most only focus on detecting a single activity of interest [26,27,28].

Deep learning techniques have been successfully applied to time series data in the healthcare domain, significantly advancing applications such as sleep staging [29,30] and stroke rehabilitation [31]. For instance, convolutional neural networks (CNNs) and recurrent neural networks (RNNs) have been utilized to automatically classify sleep stages from electroencephalogram (EEG) signals, providing high accuracy and reducing the need for manual scoring by experts [32]. These models automatically extract information-rich features and leverage the temporal dependencies and patterns inherent in time series data, enabling more accurate and efficient analysis of physiological signals [33]. Despite the widespread success of deep learning in general and in time series, the application to human activity recognition via wearable signals remains to be investigated [34].

In this investigation, we employ stringent leave-one-subject-out cross-validation (LOSO-CV), ensuring our models are tested on unseen subjects, thus avoiding any information leakage and presenting a more realistic assessment of their generalization capabilities. We notice a significant research gap in evaluation techniques for time series classification and, thus, human activity recognition; therefore, we impose LOSO-CV to provide a conservative performance estimation, ensuring that reported performance metrics reflect the model’s potential in real-world scenarios.

Despite significant innovation in methods for human activity recognition using sensor data obtained from wearables, there remains a broad research gap for the development of a system that can (1) concurrently recognize multiple human activities (2) using only accelerometer data (3) obtained via a wrist-worn smartwatch. To address this research gap, we leveraged novel techniques in deep learning and stringent evaluation criteria for concurrently recognizing various human activities, making it possible for wearable sensors to reliably characterize and monitor human behavior unobtrusively. Our system’s ability to generalize across diverse activities and settings while maintaining high accuracy demonstrates a significant advancement in the field, offering robust and scalable solutions for real-time activity monitoring.

## 2. Materials and Methods

### 2.1. Participants

Adults (n = 60) were recruited to participate in this study. Before participation, each individual was thoroughly briefed on the study’s objectives, procedures, potential risks, and benefits associated with their involvement. Following the briefing, verbal informed consent was voluntarily obtained from all participants. The consent process was conducted in accordance with ethical guidelines set forth by the institutional review board (IRB) that approved the study. The participants were assured of their right to withdraw from the study at any point without any consequences.

The demographic statistics of the participants included age, gender, and ethnicity distributions. The average age of participants was 20 years (range: 18–63 years), with a gender distribution of 87% female, 10% male, and 3% other. The ethnic composition of the sample included 82% White, 12% Black or African American, and 6% Asian. Of the participants, 87% were right-hand dominant and 13% were left-hand dominant.

To protect participant privacy and adhere to ethical standards, all data collected during the study were anonymized. Identifiable information was securely stored and accessible only to authorized personnel involved in the study, ensuring compliance with data protection regulations. This comprehensive approach to participant recruitment and management was designed to foster a respectful, ethical, and scientifically rigorous research environment.

### 2.2. Equipment

For this study, we used two models of Android smartwatches, the Ticwatch E and Ticwatch E3, that operate on Google Wear OS. The Ticwatches were chosen for their robust performance in real-time data processing and compatibility with our custom software. A custom-developed application named ASPIRE (v4.0), specifically designed for Android smartwatches to facilitate the collection of sensor data and critical for detecting and analyzing human gestures, was utilized. The software was configured to collect data continuously at a high sampling rate of 100 Hz (that is 100 Hz for each channel), sufficient for capturing the nuanced dynamics of the gestures involved in the activities targeted for the study (eating, yoga-like poses, medication-taking, and smoking simulations). Sensor data from each participant were recorded and stored directly on the smartwatch in CSV format. This method was chosen for its simplicity and effectiveness in handling large volumes of data generated at high sampling rates. The smartwatches were configured to align their timestamps with an external clock to maintain data integrity and ensure precise synchronization. This synchronization is crucial for accurately matching the collected data with the specific times of the activities recorded during the protocol. After the data collection phase, the CSV files were securely transferred to a Linux-based server for further processing, analysis, model development, and model validation. This comprehensive setup of advanced wearable technology, custom software, and meticulous data management protocols was designed to support the reliable collection of high-fidelity sensor data. These measures are critical in facilitating the accurate analysis of gesture-based activities and contribute significantly to the study’s success in advancing human activity recognition using wearable technology. The ASPIRE software is available upon request to support further research in human activity recognition.

### 2.3. Data Collection

We established a structured protocol where participants engaged in predefined activities—eating, performing yoga-like poses, simulating medication intake, and mimicking smoking—in a controlled laboratory environment. Following informed consent, participants were equipped with a smartwatch running the ASPIRE software, worn on their right (not necessarily dominant) wrist. The four predefined activities are described below.

**1. Eating Behavior**. Participants were given pizza to eat ad libitum. Participants were also given the option to simulate eating without consuming pizza for 8 min.

**2. Yoga-like Posing Behavior**. Following the eating activity, participants were guided through a sequence of yoga-like poses. This sequence included standing with arms at their sides, arms outstretched to the sides, and hands positioned atop their heads. A proctor, adhering to a standardized script, led the participants through these movements to ensure uniformity across sessions.

**3. Medication-taking Behavior**. Post-pose sequence, participants were handed empty pill bottles and instructed to simulate ‘natural’ medication-taking behaviors five times. This was followed by a ‘scripted’ sequence of medication-taking gestures, repeated five times, as outlined in a provided handout. These activities were designed to capture both spontaneous and structured movements associated with taking medication.

**4. Smoking-like Behavior**. The final activity involved participants simulating smoking using a straw, mimicking the action for at least 10 ‘puffs’. This activity aimed to replicate the gestures associated with cigarette smoking.

After these activities, participants removed the smartwatch and concluded their participation. This protocol facilitated the collection of rich accelerometer data across varied gestures and ensured that the data reflected both natural and controlled movements, critical for training our neural network model. Each activity was meticulously recorded and timestamped via the ASPIRE software, allowing for precise synchronization and analysis in subsequent stages.

### 2.4. Data Annotation

To ensure accurate identification of the start and end times of each activity, we employed a dual approach combining software-enabled self-reporting with expert post-processing. During data collection, participants utilized the ASPIRE software, which features a user-friendly interface including a drop-down menu tailored to each activity. Participants initiated and concluded each activity recording session by pressing a designated button automatically logging the timestamps and associated activity labels.

However, discrepancies can arise between self-reported timestamps and the actual start or end of activities. For example, the participant may tap “start” long before the true beginning of an activity. Such discrepancies were often evident upon review, necessitating further refinement. To address this, expert annotators meticulously examined each recorded session. Two expert annotators were chosen based on their time series analysis and pattern recognition experience. To prepare the annotators for this labeling task, each was given a smartwatch with the ASPIRE software, the data labeling software, and the structured protocol. Annotators were proctored through the protocol, self-labeling the start and end of each activity, and then asked to familiarize themselves with the spatial characteristics of acceleration signals for each gesture.

After expert annotators were trained, one was considered the principal annotator and the other secondary. The principal annotator was provided with each of the 60 recordings and the self-reported labels. Recordings and labels were visualized using our custom data visualization and labeling software. During visualization, the principal annotator assessed and adjusted the self-reported beginning and end of each gesture based on the corresponding spatial characteristics of the acceleration signal. After the principal annotator completed their evaluation of each recording, the secondary annotator visualized the accelerometer signals and corresponding annotations as adjusted by the principal. The secondary annotator reviewed all annotations, biasing the principal’s decision given a discrepancy to resolve disagreements. However, we noticed that all discrepancies were less than a quarter of a second and therefore hypothesized that the difference would not change the result of training.

Expert evaluation of self-reported labels was critical in maintaining the integrity and reliability of our dataset, facilitating robust analysis during the model training phase.

### 2.5. Model Design

Human activity data contain various spatial and temporal features. To automatically extract information-rich features for downstream classification, we leveraged a hybrid deep learning architecture that combines the strengths of convolutional neural networks (CNNs) and recurrent neural networks (RNNs). Residual Networks (ResNets) [35] are well known to handle spatial hierarchies and have become nearly synonymous with CNNs, given the seemingly necessary use of residual connections in CNNs. Specifically, our model incorporates a RegNet-style [36] 1D ResNet, shown to have a smaller design space of hyperparameters that lead to monotonically better performance. The convolutional encoder is followed by a Long Short-Term Memory (LSTM) network [37]. This architecture was chosen to capitalize on the ability of RegNet to efficiently handle spatial hierarchies and the ability of LSTM to manage time-dependent sequences, making our hybrid architecture particularly suited for interpreting complex human activities through accelerometer data.

First, the 3-channel 100 Hz raw acceleration signals are input to the RegNet, which encodes input time series into a high-dimensional feature space. We pretrain the RegNet using a linear layer and labeled examples. After pretraining the encoder, we freeze encoder weights, remove the linear layer, and train the LSTM component. For all components of our model, we train and evaluate using only the dataset described above (i.e., some train and test split of the 60 recordings obtained and annotated).

#### 2.5.1. Model Training Process


**Pretraining Convolutional Encoders:**
Initially, the convolutional encoders, modeled after the RegNet architecture but adapted for 1D input, were trained to extract relevant features from the raw accelerometer data. These encoders were designed to identify subtle patterns and characteristics that distinguish between human activities.To optimize the parameters of the convolutional encoder, random mini-batches of 512 samples were selected, where each sample was a window of 3-channel accelerometer data with 1001 points (i.e., approximately 10 s) and the corresponding activity annotation.Once adequately trained, these convolutional encoders were frozen. This means their weights were fixed, preventing them from updating during the subsequent training phase. Freezing the encoders helps transfer learned features and stabilizes the training of the LSTM layers.
**Training LSTM Networks:**
Following the pretraining of the CNN layers, the LSTM component of the model was trained. The LSTM layers are designed for sequential data, making them ideal for interpreting the temporal dependencies and dynamics captured by RegNet.To optimize the parameters of the LSTM component of the model, random mini-batches of 32 samples were selected, where each sample was a sequence of 131 001 sample windows of 3-channel accelerometer data (i.e., approximately 130 s) and the corresponding activity annotation from the center window in the sequence.This sequential training approach, where the LSTM layers are trained after the CNN layers have been frozen, ensures that our model can identify relevant features and analyze their progression over time.

#### 2.5.2. Optimization and Hardware Utilization

The models were optimized using the AdamW [38] optimizer, an adaptation of the traditional Adam [39] optimizer incorporating a decoupled weight decay regularization. Decoupled weight decay helps in improving training stability and convergence. Gradient descent updates were efficiently distributed across two NVIDIA GeForce RTX 4090 GPUs with PyTorch [40]. Utilizing these high-performance GPUs allowed us to handle the substantial computational demands of training deep neural networks, significantly speeding up the process and enhancing model performance through parallel processing. This implementation strategy ensures efficient learning and maximizes the potential of our neural network to generalize well across new, unseen data, crucial for deploying the model in real-world applications where diverse human activities need to be accurately recognized and interpreted.

### 2.6. Validation Paradigm

The validation paradigm is the cornerstone of model evaluation. In the context of our study, which utilizes subject-based datasets, there is a high risk of information leakage, where the model might inadvertently learn details specific to individual subjects in the training set, rather than the underlying patterns applicable across different individuals. To mitigate this risk and encourage true generalization, we employ a stringent cross-validation approach.

We adopt the leave-one-subject-out cross-validation (LOSO-CV) method, particularly well suited for our dataset with 60 subjects, each providing 30 min of recorded data. In this paradigm, each subject’s dataset forms a single fold. We train the model on data from 59 subjects and then test it on the remaining one. This process is repeated such that each subject is used exactly once as the test set. This method of validation is rigorous and aligns with our goal to ensure that the model can generalize across different subjects without bias.

This LOSOCV approach helps validate the effectiveness of the model under conditions of data scarcity and provides a reliable measure of how well the model can be expected to perform in practical settings, where it will encounter data from new individuals. Such a robust validation framework is crucial for advancing machine learning applications in wearable technology, particularly for interpreting complex human behaviors through sensor data.

### 2.7. Evaluation Metrics

Defining metrics for evaluating deep learning models is critical for establishing proper results. Cross entropy loss is a good and robust metric for classification [41,42,43,44]; however, it is difficult to interpret. Also, a common mistake in evaluating an imbalanced classification problem is the use of accuracy as a performance metric. It is easy for a deep learning model to learn class imbalance through mini-batches and bias predictions after that. To avoid artificially high accuracy due to bias, we define two goals that we want to maximize. In terms of activity recognition, we want to predict as many of the true activities as possible correctly. We also want to minimize the number of incorrectly predicted activities. These two requirements equate to two popular metrics recall and precision:$$\begin{array}{cc} \mathrm{Precision} & =\frac{\mathrm{TP}}{\mathrm{TP}+\mathrm{FP}}\\ \mathrm{Recall} & =\frac{\mathrm{TP}}{\mathrm{TP}+\mathrm{FN}}\; \end{array}$$

Precision and recall can be computed from a confusion matrix. To combine precision and recall into a single reportable metric, we define the *F*_1_-score, the harmonic mean between precision and recall:$$F_{1}=2\cdot \frac{\;\mathrm{precision}\;\cdot \;\mathrm{recall}\;}{\;\mathrm{precision}\;+\;\mathrm{recall}\;}=\frac{\mathrm{TP}}{\mathrm{TP}+\frac{1}{2}\left(\mathrm{FP}+\mathrm{FN}\right)}$$

The *F*_1_ score is one of the most commonly used metrics in deep learning scenarios, and ranges from 0 to 1, similar to accuracy, where 1 is a perfect classifier [45].

To evaluate the efficacy of our hybrid neural network model in classifying five distinct human activities—other (anything except the 4 remaining behaviors), eating, exercise, medication, and smoking—based on accelerometer data, we designed a comprehensive set of visualizations that describe various aspects of the model’s performance. A recall confusion matrix was generated to provide a detailed breakdown of the model’s predictive accuracy across classes, facilitating an assessment of precision for each activity type. A confusion matrix shows which activities are most reliably detected and which are likely to be confused.

Furthermore, to explore the relationship between the model’s confidence in its predictions and its actual performance, we plotted the macro *F*_1_ scores against the average network confidence. This scatter plot, enhanced with a Pearson correlation coefficient, serves to quantify the consistency of the model’s confidence with the accuracy of its predictions, providing a metric of reliability.

Additionally, we employed box plots to depict the distribution of key performance metrics—confidence, *F*_1_ score, precision, and recall—across cross-validation folds. These plots are critical for visualizing the variability and robustness of the model across different activity classifications, highlighting any significant outliers or inconsistencies.

Lastly, the influence of the confidence threshold on the model’s overall performance was analyzed through a curve plotting the average *F*_1_ score against varying confidence thresholds. This performance curve is instrumental in determining the optimal threshold that balances sensitivity and specificity, which is crucial for practical applications in health monitoring and behavior analysis.

Each of these visualizations plays a pivotal role in providing a holistic understanding of the model’s capabilities and limitations, thereby guiding further refinement and application of the classifier in real-world settings.

## 3. Results

### 3.1. Dataset Qualitative Results

Our data collection resulted in exactly 1 recording for each of 60 participants. Sensor data were collected at 100 Hz. Each recording consisted of a timestamp and a 3-dimensional acceleration vector for each sample. On average, each participant produced 30 min of sensor data (i.e., approximately 180,000 samples). Thus, we aggregated approximately 11 million data points or 30 h of human activity data sampled at 100 Hz.

### 3.2. Segmentation Performance

In this study, model performance was meticulously evaluated across 60 folds, corresponding to the leave-one-subject-out cross-validation framework, where each fold represents data from a unique participant. This approach allowed us to assess the model’s capability to accurately segment and classify activity types within entire recordings from a hold-out participant not included in the training data.

The aggregated results are summarized in Table 1. The average macro *F*_1_ score across all folds was 0.851, indicating a high level of model performance in segmenting and recognizing the specified activities. The precision and recall were also robust, with scores of 0.856 and 0.871, respectively. These results suggest a well-balanced model that performs reliably across individuals without significant bias toward any particular activity class.

The observed balance between recall and precision around the *F*_1_ score is likely attributable to the moderate class balance achieved through our data collection and model training processes. The consistency in the standard deviations (0.09 for *F*_1_, 0.08 for precision, and 0.08 for recall) across the metrics further illustrates the model’s stability and the reliability of our validation paradigm.

The results depicted in Figure 1 panels collectively highlight the strengths and areas for improvement in our hybrid neural network model’s ability to classify five distinct activities from accelerometer data. The confusion matrix in the top-left panel indicates a high recall for the ‘exercise’ and ‘medication’ categories, suggesting that the model is particularly effective in identifying periods of no activity and medication-related actions. However, the lower recall values for ‘smoking’ and ‘eating’ suggest these activities are more challenging for the model, potentially due to similar movement patterns or insufficient training data.

The top-right panel further substantiates the model’s overall reliability as evidenced by a strong positive correlation (r = 0.71) between average network confidence and macro *F*_1_ score. This correlation confirms that higher confidence in predictions generally translates to higher accuracy, reinforcing the model’s utility in real-world applications where confidence thresholds could be adjusted to optimize performance.

In the bottom-left panel, the box plots reveal a broad spread in the precision and recall metrics for different activities, particularly ‘Smoking’, which exhibits lower recall. This variability underscores the need for targeted improvements in the model’s training regimen, perhaps by incorporating more diverse data samples for underperforming categories.

Lastly, the performance curve in the bottom-right panel illustrates a significant increase in the average *F*_1_ score as the confidence threshold is raised, peaking before it plateaus. This suggests that setting an optimal confidence threshold could dramatically enhance the model’s precision without substantially sacrificing recall, which is crucial for deploying the model in scenarios where accuracy is paramount.

Together, these results provide valuable insights into the model’s operational characteristics and offer clear directions for further refining its performance, ensuring that it can be effectively tailored to specific requirements of activity-based classifications in health monitoring systems.

Figure 2 provides a detailed depiction of the model’s performance in a best-case scenario for activity classification over a significant data range. The top panel shows three signal plots (green, orange, and blue) illustrating the raw sensor outputs over time. These plots highlight the variability and complexity of the data that the model must interpret. The second panel from the top displays the reference or ground truth for different activity states, marked distinctly by changes in the level. Correspondingly, the third panel shows the model’s predictions, closely mirroring the reference signals, indicating a high degree of accuracy in activity classification.

Most notably, the bottom panel presents a probabilistic view of the model’s confidence across different classes, represented by color-coded regions (purple, blue, green, beige, and red), with the model’s confidence peaking in regions where the predictions and references align closely. This visual correlation between high-confidence areas and accurate predictions underscores the model’s effectiveness in distinguishing between various activities under optimal conditions.

The precision in tracking the reference signal and the consistency in high-confidence predictions highlight the model’s robustness and reliability, which are crucial for practical applications in real-time activity monitoring. This result suggests that the model not only performs well under controlled conditions but also demonstrates the potential to operate effectively in dynamic, real-world environments, where accurate and reliable activity recognition is essential.

Figure 3 illustrates the model’s performance in an average case scenario, based on data aggregated across 60 folds with an achieved macro *F*_1_ score of approximately 0.85. The top panel displays the raw 100 Hz acceleration signal, which serves as the sole input for the neural network, highlighting the raw data’s complexity. The second-from-top panel shows the reference label signal defined by experts, with labels corresponding to activities such as resting, eating, exercising, taking medication, and smoking. The third-from-top presents the predicted labels from the neural network, where notable deviations from the reference signal indicate areas where the model’s performance could be improved.

Most critically, the bottom panel visualizes the probability distribution of the network’s predictions over time, color-coded to reflect confidence levels associated with each predicted activity class. The regions where the predicted and reference labels match demonstrate areas of higher confidence, while mismatches are characterized by lower confidence, illustrating the model’s uncertainty in these segments. This visualization not only corroborates the model’s capability to accurately recognize and classify activities under typical conditions but also pinpoints the specific contexts in which the model struggles, offering a clear direction for focused improvements in future iterations. These insights are crucial for refining the model to enhance its reliability and applicability in diverse real-world monitoring tasks.

With the intent of the trained model being deployed on wearable devices, the evaluation of computational efficiency is critical. The purpose of this investigation was not to investigate computational efficiency empirically; however, we provide a brief discussion here. Given the prediction window is approximately 10 s long, a wearable must make an inference once every 10 s. Given that two adjacent prediction sequences share 12 of the same convolutional embeddings, we can cache 12 embeddings and forward propagate one convolutional embedding and one LSTM inference every 10 s. Anecdotally, we have found this is easily achievable on commercially available smartwatches alongside the ASPIRE software and the effect on battery life is negligible.

## 4. Discussion

This study has proposed a single unifying model for concurrently identifying human activities. Commonly, machine learning practitioners improperly validate model performance leading to overoptimistic results. Here, we have provided a rigorous leave-one-subject-out cross-validation paradigm and therefore a conservative estimate of performance. We have achieved an average macro *F*_1_ score of approximately 0.85 across 60 folds of cross-validation. *F*_1_ score in the context of segmentation of time series is proportional to the amount of overlap between predicted and reference regions for a given activity. In certain use cases, an exact segmentation of a given human activity signal is not necessary. One limitation of this work is the potential for the model to learn the sequence of activities due to the structured protocol, which could be mitigated by randomizing the sequence of activities.

Passive eating detection systems have previously been tested solely for their accuracy in detecting eating or drinking events [9,10,11]. The current model, however, captures complex human activities comprehensively and sequentially, providing numerous implications for improving diet assessment as well as behavioral interventions. By implementing the current system in free-living settings, it is possible to identify patterns of activity that occur before other activities. For example, drinking alcohol might be found to be a reliable predictor of smoking events. Another example would be overeating or smoking more frequently on days when medication was not taken, all of which could be detected and quantified passively and remotely. Identifying the antecedents and precedents of each individual’s activity is beneficial to better understand their eating behavior patterns and to create individualized messages to target specific behaviors [46]. Moreover, most of the current image-based approaches require participants to take photos of meals and snacks before and after consumption to maximize the accuracy of dietary assessments [47]. This emphasizes the immense potential for integrating eating detection systems with image-based methods, which can then be used to prompt participants to record what they eat and contextual information about the eating event. Therefore, the developed model has the potential to serve as a powerful tool to supplement image-based dietary assessment tools by providing comprehensive information on eating behavior, thereby enhancing dietary compliance.

Similarly, for medication-taking behaviors, technologies such as medication event monitoring systems exist, providing real-time detection of when a medication bottle is opened [48]. However, they do not detect if a medication was removed from the bottle or whether the medicine seemed to have been placed in the person’s mouth, both of which could be detected by our system. In addition, our system could be programmed to know the time when a person should take their medications, and if the motions of medication-taking have not been detected, a reminder could be sent. In this manner, it could support medication adherence.

Our system’s ability to generalize across diverse activities and settings while maintaining high accuracy demonstrates a significant advancement in the field, offering robust and scalable solutions for real-time activity monitoring. Therefore, the developed model has the potential to serve as a powerful tool for the characterization of human behavior in real-world contexts and contribute to developing a comprehensive model of human health. Collectively, we have demonstrated the potential of AI and its contributing role to healthcare during the early phases of diagnosis, prognosis, and/or intervention. The proposed model improves the accuracy of activity detection and offers valuable promise for personalized interventions, ultimately contributing to better health outcomes.

## Figures and Tables

**Figure 1 sensors-24-04542-f001:**
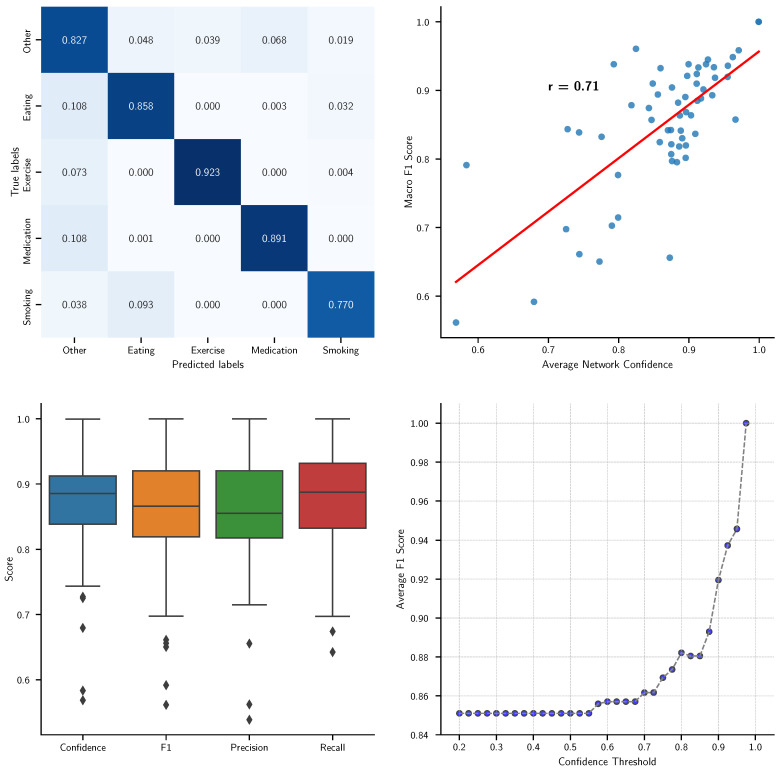
Comprehensive evaluation of a hybrid neural network model for classifying five types of activities from accelerometer data. **Top-left:** Confusion matrix displaying the recall of model classifications for activities (‘other’, ‘eating’, ‘exercise’, ‘medication’, and ‘smoking’). **Top-right:** Scatter plot of the macro *F*_1_ score versus average network confidence, with a Pearson correlation coefficient (r = 0.71) indicated by the red line. **Bottom-left:** Box plots showing the distribution of confidence, *F*_1_ score, precision, and recall metrics across cross-validation folds, illustrating performance consistency and variability. **Bottom-right:** Curve illustrating the relationship between the confidence threshold and the average *F*_1_ score, demonstrating how model performance optimizes at higher thresholds. These panels collectively highlight the model’s effectiveness and potential utility in real-world applications for health monitoring and behavior analysis.

**Figure 2 sensors-24-04542-f002:**
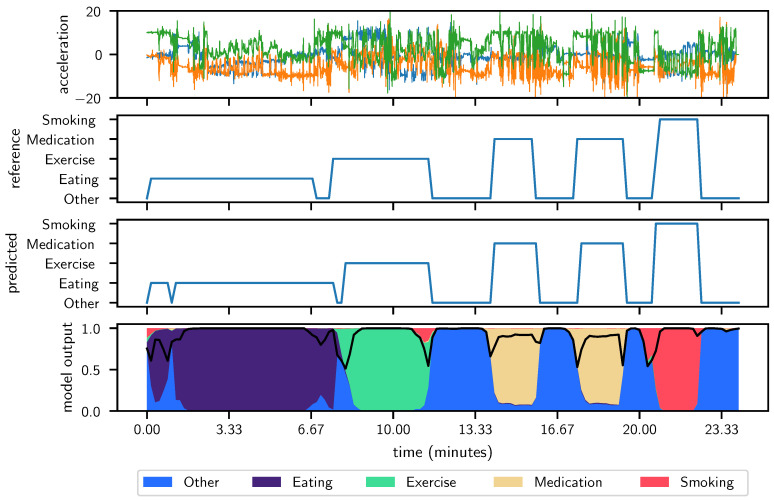
Best case participant across 60 folds, achieving *F*_1_ score of approximately 0.98. The shared x-axis between all panels represents the data-point index in time at 100 Hz. **Top panel:** Raw 100 Hz acceleration signal used as sole input to our deep neural network. **Second-from-top panel:** Reference label signal defined by experts where labels 0 through 4 are mapped to other, eating, exercise, medication taking, and smoking, respectively. **Third-from-top panel:** Predicted label signal as output directly from our deep neural network. The goal is to produce a label signal that looks identical to the reference signal. **Bottom panel:** The probability distribution of the output of the deep neural network. Legend to be added. The sum of probability for all classes sum to one. At each point, the maximum probability class is taken to be the true class in the predicted label signal. The black line shown on the probability distribution is the confidence at each point in time. Confidence is the magnitude of the probability of the class with the greatest probability. Lower confidence indicates closer to a random guess by the network. Higher confidence indicates that the network is “almost surely” correct.

**Figure 3 sensors-24-04542-f003:**
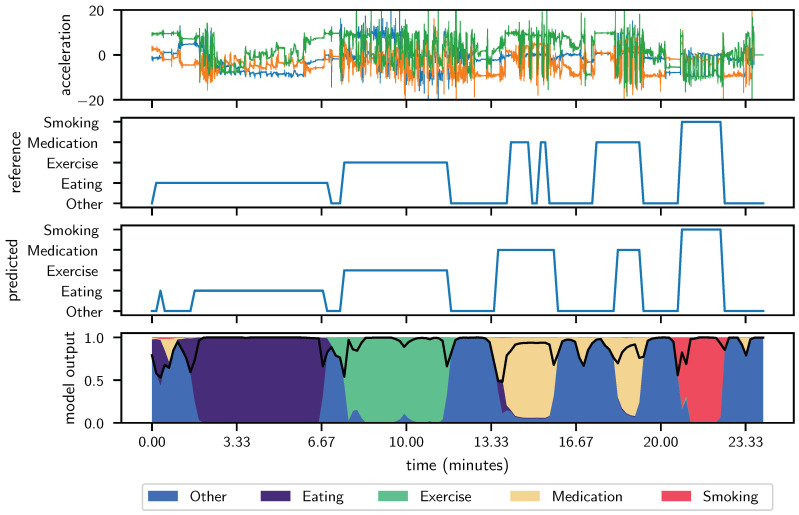
Average case participant across 60 folds achieving *F*_1_ score of approximately 0.85. The shared x-axis between all panels represents the data-point index in time at 100 Hz. **Top panel:** Raw 100 Hz acceleration signal used as sole input to our deep neural network. **Second-from-top panel:** Reference label signal as defined by experts where labels 0 through 4 are mapped to other, eating, exercise, medication taking, and smoking, respectively. **Third-from-top panel** Predicted label signal as output directly from our deep neural network. The goal is to produce a label signal that looks identical to the reference signal. **Bottom panel:** The probability distribution of the output of the deep neural network. Legend to be added. The sum of probability for all classes sum to one. At each point, the maximum probability class is taken to be the true class in the predicted label signal. The black line shown on the probability distribution is the confidence at each point in time. Confidence is the magnitude of the probability of the class with the greatest probability. Lower confidence indicates a closer to random guess by the network. Higher confidence indicates that the network is “almost surely” correct.

**Table 1 sensors-24-04542-t001:** Summary of performance metrics for a five-activity classification model. The table presents the *F*_1_ score, precision, recall, and support for each activity class—other, eating, exercise, medication, and smoking—over multiple folds of the data. The ‘macro avg’ represents the average performance metrics computed by averaging the scores of all classes, treating all classes equally. The ‘weighted avg’ provides an average score weighted by the support (the number of true instances for each label), reflecting the influence of each class on the overall metric. These statistics demonstrate the model’s balanced performance across different types of activities, highlighting its potential applicability in diverse real-world scenarios.

Activity	*F* _1_	Precision	Recall	Support
Other	0.87	0.83	0.84	75.02
Eating	0.88	0.86	0.85	33.05
Exercise	0.86	0.92	0.88	16.45
Medication	0.84	0.89	0.86	25.02
Smoking	0.82	0.86	0.82	16.48
Macro Avg.	0.84	0.85	0.83	164.37
Weighted Avg.	0.87	0.86	0.85	164.37

## Data Availability

The data presented in this study are available on request from the corresponding author.

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
