# Peer review of "Toward Concurrent Identification of Human Activities with a Single Unifying Neural Network Classification: First Step"

_sensors, 2024, doi:10.3390/s24144542_

Round 1

Reviewer 1 Report

Comments and Suggestions for Authors

Thank you for contributing an interesting and well-written paper. There
are a few minor point that would benefit from further clarification. And
there is one issue that needs to be explained in more detail, namely
that of the expert annotations. Please find below a list of these
issues, and some ideas of how you might resolve them.

Materials and Methods
═════════════════════

ASPIRE software
───────────────

  • Can you provide a reference or link to further information about the
    ASPIRE software. Other researchers might be interested in using the
    software in their own research.

Lines 176–182: Expert annotations
─────────────────────────────────

  • The annotation process needs to be described in more detail:
    • How many expert annotators were there, and what qualified them as
      expert annotators?
    • What instructions/guidance was given to annotators?
    • How did the annotators examine each session? What aspects/data did
      they examine? Did they have access to the self-reported activity
      logs?
    • How did they record their annotations? E.g., correcting
      self-annotations in a graphical user interface, recording
      start/end times in a spreadsheet while viewing a video, …?
    • How did you resolve disagreements between annotators? How much
      disagreement was there?

Validation Paradigm
═══════════════════

LOSO-CV and Information Leakage
───────────────────────────────

  The LOSO-CV protects against the model overfitting to a specific
  person. However, it is possible that the model learned (parts of) your
  protocol. I.e., the model might simply have learned that people:
  1. First eat (for about 8 minutes), then
  2. do yoga (same sequence of poses, same standardized script → similar
     duration), then
  3. take medication, and finally
  4. smoke about 10 puffs.
  Randomizing the sequence of activities would go a long way toward
  mitigating against this.

  I think this is a limitation of the study design that you ought to be
  mentioned in the discussion.

Results
═══════

Segmentation Performance
────────────────────────

  I presume that the “Other” and “None” activity both refer to the same
  thing: Any activity other than “Eating” “Exercise” “Medication” or
  “Smoking”. If so, please use one term (I’d use “Other”) throughout. If
  not, please explain how they are different.

Figures 2 and 3
───────────────

  Those legends do need to be added.

Reviewer 2 Report

Comments and Suggestions for Authors

It is an interesting research, however, there are several aspects that require further elucidation to enhance the clarity and robustness of the findings:

1. Missing legend in Figures 2 and 3: The visual representation of data in Figures 2 and 3 is good; however, the absence of a legend for the probability distribution derived from the deep neural network's output is a notable oversight. A comprehensive legend would not only aid in the interpretation of these figures but also strengthen the validity of the visual data presented. 

2. Data Categorization Methodology: The use of the Ticwatch to record a variety of activities, such as eating, performing yoga-like poses, taking medication, and smoking, is innovative. However, the methodology by which experts analyzed the data to ensure accurate categorization of these activities is not explicitly described. Providing a clear description of the benchmark method or algorithm used to differentiate between these actions would greatly benefit the transparency and replicability of the study.

3. Sampling Frequency Clarification: The study mentions a sampling frequency of 100 Hz. Given that there are three channels of data acquisition, it is crucial to clarify whether this frequency applies to each channel individually or collectively. If the sampling frequency is indeed divided among the channels, stating that the effective sampling rate per channel is one-third of 100 Hz would be necessary to avoid any misconceptions regarding the data resolution.

4.  Model Training Details: The convolutional neural networks (CNNs) and long short-term memory (LSTM) networks used in the study are powerful tools for pattern recognition and time-series analysis, respectively. To better understand the training process, it would be helpful to know the exact number of datasets utilized for training these models. Additionally, insights into the computational efficiency of the trained models, such as the time taken to classify the four activities, would provide a more comprehensive view of the model's performance in practical applications.

Reviewer 3 Report

Comments and Suggestions for Authors

The authors propose a model for concurrently identifying human activities. The topic of the study is relevant. The methods are based on neural networks, and the results can be interesting for researchers and healthcare specialists. The analysis of the results performed very well.

Minor comments

1. Please identify in the introduction what research gap overcome this work.

2. Section 2.1 requires more statistics related to the characteristics of the participants.

3. The choice of the neural architecture is not explained.

Round 2

Reviewer 3 Report

Comments and Suggestions for Authors

All my comments from the first review have been carefully addressed. I suggest accepting this paper in the case of consent of the editors.